# Structural and Thermodynamic Insights into Dimerization Interfaces of Drosophila Glutathione Transferases

**DOI:** 10.3390/biom14070758

**Published:** 2024-06-26

**Authors:** Mathieu Schwartz, Nicolas Petiot, Jeanne Chaloyard, Véronique Senty-Segault, Frédéric Lirussi, Patrick Senet, Adrien Nicolai, Jean-Marie Heydel, Francis Canon, Sanjiv Sonkaria, Varsha Khare, Claude Didierjean, Fabrice Neiers

**Affiliations:** 1Flavour Perception: Molecular Mechanisms (Flavours), INRAE, CNRS, Université de Bourgogne, 21000 Dijon, France; mathieu.schwartz@inrae.fr (M.S.); jeanne.chaloyard@u-bourgogne.fr (J.C.); veronique.senty-segault@inrae.fr (V.S.-S.); jean-marie.heydel@u-bourgogne.fr (J.-M.H.); 2Laboratoire Interdisciplinaire Carnot de Bourgogne, UMR 6303 CNRS, Université de Bourgogne Franche-Comté, 21078 Dijon, France; nicolas.petiot01@u-bourgogne.fr (N.P.); patrick.senet@u-bourgogne.fr (P.S.); adrien.nicolai@u-bourgogne.fr (A.N.); 3UMR 1231, Lipides Nutrition Cancer, INSERM, 21000 Dijon, France; frederic.lirussi@univ-fcomte.fr; 4UFR des Sciences de Santé, Université de Bourgogne Franche-Comté, 25000 Besançon, France; 5Plateforme PACE, Laboratoire de Pharmacologie-Toxicologie, Centre Hospitalo-Universitaire Besançon, 25000 Besançon, France; 6Independent Researcher, 21000 Dijon, France; 7Soft Foundry Institute, Seoul National University, Kwanak-gu, Seoul 39-131, Republic of Korea; ssonkaria64@snu.ac.kr (S.S.); khare@snu.ac.kr (V.K.); 8CRM2 Laboratory, CNRS, Université de Lorraine, 54000 Nancy, France; claude.didierjean@univ-lorraine.fr

**Keywords:** GST, glutathione transferases, dimerization interface, protein stability, *Drosophila melanogaster*, Delta GST, Epsilon GST

## Abstract

This study presents a comprehensive analysis of the dimerization interfaces of fly GSTs through sequence alignment. Our investigation revealed GSTE1 as a particularly intriguing target, providing valuable insights into the variations within Delta and Epsilon GST interfaces. The X-ray structure of GSTE1 was determined, unveiling remarkable thermal stability and a distinctive dimerization interface. Utilizing circular dichroism, we assessed the thermal stability of GSTE1 and other Drosophila GSTs with resolved X-ray structures. The subsequent examination of GST dimer stability correlated with the dimerization interface supported by findings from X-ray structural analysis and thermal stability measurements. Our discussion extends to the broader context of GST dimer interfaces, offering a generalized perspective on their stability. This research enhances our understanding of the structural and thermodynamic aspects of GST dimerization, contributing valuable insights to the field.

## 1. Introduction 

Glutathione transferases (GSTs) are enzymes found universally across all kingdoms of life, spanning more than six decades of research [1]. In insects, they play pivotal roles in various processes, including detoxification, ecdysteroid biosynthesis, or chemoperception [2,3,4,5,6]. The GSTome of an organism including all the enzymes catalyzing the glutathione transferase activity is categorized into three main superfamilies based on their cellular localization: mitochondrial, microsomal, and cytosolic (also named canonical GSTs) [7]. While the mitochondrial class (Kappa GSTs) is widespread among eukaryotes, it is notably absent in insects. The microsomal GST (also named MAPEG) is not common in insects; for example, *Drosophila melanogaster* possesses only one microsomal GST. The cytosolic or canonical GSTs have evolved from a common ancestor and represent the largest family. GSTs are denoted by Greek letters for each class potentially containing multiple GSTs, which are designated by Arabic numerals. The number of cytosolic GSTs varies across species in a general manner primarily due to number differences for a specific class. For insects, the Delta and Epsilon classes’ occurrences are the major driver in this number difference [8]. Consequently, these two classes can be considered as an evolutionary lever to support insect adaptation. Interestingly, Delta and Epsilon classes are specifically found in arthropods and are represented by 25 members out of the total 42 cytosolic GSTs in *D. melanogaster*. Phylogenetic analyses of the protein sequences of Delta and Epsilon GSTs indicate two distinct groups, each corresponding to one of these classes [9]. Within the 14 Epsilon GSTs there are three subgroups: one resulting from an ancient separation, which includes GSTE11, GSTE12, GSTE13, and GSTE14, while the other two groups emerged in a subsequent period of time with one group comprising GSTE1, GSTE2, GSTE9, and GSTE10 and the remaining belonging to the last group. GSTs play a crucial role in conferring adaptation to various xenobiotics by metabolizing or sequestering these chemicals and, consequently, providing a protection against them. The structural diversity of GSTs, particularly within their enzymatic active sites, is key to their ability to perform a wide range of functions [10] and to catalyze a diverse range of substrates. GSTs were categorized into four types based on the catalytic amino acids, namely, tyrosine (TyrGSTs), serine (SerGSTs) including the Delta and Epsilon classes, cysteine (CysGSTs), and atypical (AtyGSTs) [11]. Among the canonical GSTs, the majority of the classes are dimeric with few exceptions such as the monomeric Lambda class found specifically in plants [12,13]. All the GSTs’ classes identified within the insects are recognized as dimeric including the Delta and Epsilon. For the dimeric GSTs, the dimerization interface is important to support their activity [14]. The affinity of both subunits is generally low, and evaluation of GSTP1, for example, in the nM range supports the dimeric state under physiological conditions [15]. Additionally, the dimeric state of GSTs supports allosteric properties including positive and negative cooperativity [16,17]. The evolution of large multigenic classes such as the Delta and Epsilon found in insects allows introspection to extend the capacity to catalyze a large number of chemical families and further permits the evolution of certain GSTs toward new functions thanks to the redundant functions observed for the GSTs. Insect GSTs are involved in chemical detoxification, oxidative stress resistance, and some more specific functions in *D. melanogaster* GSTE14, which was shown to be involved in ecdysteroid biosynthesis and likely supported by an isomerase activity [18,19]. Additionally, the GSTs’ functions directly correlate to their location supported by specific and distinct tissue expression patterns for each GST [9].

The diversity of functions, substrates, and locations of GSTs is also observed at the level of the dimerization interface. Different types of motifs are characterized depending on the class but also within the same class as the original dimerization motif observed in the unique member of the Delta class in *Apis mellifera*, which operates through a sulfur sandwich organization [20]. Clasp, wafer, and sulfur sandwich motifs of dimerization were previously observed for the Delta and Epsilon classes. In this work, based on sequence alignment, an original dimerization motif not observed to date for the GSTs was identified for GSTE1. GSTE1 was recombinantly produced, purified, and enzymatically characterized and its X-ray structure was solved. The recent appearance in an evolutionary perspective of this motif will be discussed in light of the thermal stability supported by this motif in comparison to all other motifs previously identified in *D. melanogaster* GSTs structurally characterized.

## 2. Materials and Methods

### 2.1. Cloning, Expression, and Purification of GSTs

The DNA sequences encoding *Drosophila melanogaster* GSTE1 (DmGSTE1), *D. mel* GSTE7 (DmGSTE7), *D. mel* GSTD1 (DmGSTD1), and *D. mel* GSTD10 (DmGSTD10), corresponding to the protein sequences annotated with the Q7KK90, A1ZB72, P20432, and Q9VGA1 identifiers on Uniprot.org, were codon-optimized for expression in *Escherichia coli*. The DmGSTD10 was added to a DNA sequence coding for six His at the N-terminal part of the protein. The DNA sequence synthesized by GENEWIZ (Leipzig, Germany) was subcloned into the pET22b vector except for the sequence coding GSTE1 in pET28a (Novagen, Darmstadt, Germany) between the *NdeI* and *SacI* restriction sites. Following transformation, isolated colonies from an LB-ampicillin agar plate (50 mg/L ampicillin or 50 mg/L kanamycin for the pET28a in all the following steps) were utilized to inoculate a 50 mL LB-ampicillin culture (50 mg/L ampicillin) incubated overnight at 37 °C. Subsequently, this 50 mL culture served to inoculate 4 L of LB-ampicillin medium. Protein expression induction occurred at OD 600 nm = 0.6, using a final concentration of 1 mM isopropyl β-D-1-thiogalactopyranoside (IPTG), and the growth was maintained for 18 h at 37 °C. Cells were harvested via centrifugation (4000 g, 20 min, 4 °C), followed by resuspension in 10 mM phosphate, 150 mM NaCl buffer at pH 7.0 (PBS buffer) (for GSTE1 and GSTD1), in Tris 50 mM, pH 8.0 buffer (for GSTE7), or resuspended in 20 mM sodium phosphate, 0.5 M NaCl, and 20 mM imidazole, pH 7.4, and subsequent disruption at 4 °C through sonication (Vibracell, Bioblock, Pittsburgh, PA, USA). Following centrifugation at 20,000 g for 45 min at 4 °C, the resulting supernatant was applied onto a GSTrap Fast Flow 5 mL column (DmGSTE1 and DmGSTD1), a 20 mL home packed Q sepharose Fast Flow column (DmGSTE7), or a 5 mL Ni 2+ Sepharose 6 Fast Flow column (DmGSTD10). All resins were purchased from Cytiva (Washington, DC, USA). After washing the column with the buffer corresponding to that used for cell resuspension, elution of DmGSTE1 and DmGSTD1, as well as DmGSTE7 and DmGSTD10, was performed in PBS supplemented with 10 mM of reduced glutathione (GSH), a Tris buffer containing 50 mM Tris and 1M KCl at pH 8.0, and a PBS buffer containing 0.5 M NaCl and 500 mM imidazole at pH 7.4. DmGSTD2 and DmGSTE14 were produced and purified as previously described [19,21]. Fractions containing GSTs were pooled and dialyzed against 20 mM Tris-HCl at pH 8.0 (for DmGSTE1 crystallization) or 100 mM potassium phosphate buffer at pH 6.5 (for DmGSTE1 enzymatic assays or circular dichroism experiment for all other GSTs). The purity of the protein was confirmed via Coomassie-stained 12% SDS-PAGE gel analysis.

### 2.2. Kinetic Measurements

The 1-chloro-2,4-dinitrobenzene (CDNB) and glutathione were purchased from Sigma-Aldrich (St. Louis, MO, USA). CDNB was dissolved in ethanol. The final solvent concentration in the reaction system was kept at a maximum of 5% (*v*/*v*) in a 100 mM sodium phosphate buffer at pH 6.5. Specific activities were determined spectrophotometrically at 25 °C on a Shimadzu UV-1800 spectrophotometer (Kyoto, Japan). Glutathionylation of CDNB was monitored at 340 nm (ε = 9600 M^−1^cm^−1^). Steady-state kinetic parameters were determined under the corresponding assay conditions using variable substrate concentrations and a 2 mM saturating concentration of GSH for the K_M_ of CDNB determination and using a 500 µM saturating concentration of CDNB for the K_M_ of GSH determination. The Michaelis–Menten equation was fitted to the data to determine K_M_ and V_max_. V_max_ values were transformed into k_cat_ based on the 50 nM of DmGSTE1 used for the assays.

The thiol transferase activity was measured through a glutathione reductase-coupled method, following the consumption of NADPH resulting in an absorbance decrease at 340 nm (e = 6220 M^−1^cm^−1^). The reaction mixture contained 2 mM GSH, 100 μM NADPH, 0.5 units of yeast glutathione reductase, 100 nM DmGSTE1, and 200 µM of hydroxyethyl disulfide (HED) in 100 mM Tris-HCl pH 8.0 buffer.

### 2.3. Far-UV Circular Dichroism

Far-UV circular dichroism (CD) spectra were acquired utilizing a JASCO J-815 spectropolarimeter (Tokyo, Japan) equipped with a Peltier temperature control. Proteins were prepared at a concentration of 2.5 µM in 100 mM phosphate buffer at pH 6.5. Employing a 1 cm path-length quartz cell, the protein sample spectra were collected at a scan speed of 100 nm.min^−1^ over the range of 200 to 240 nm. Spectra were averaged over three scans. Each enzyme’s spectra were consistently recorded in the same quartz cell while incrementally raising the temperature from 30 °C to 90 °C at 1 °C intervals. The data obtained at 220 nm were plotted against temperature for each enzyme. Thermal denaturation data were processed using cdpal software (version 2.18) [22]. Data were normalized and fitted to the two-state model using the standard Autofit procedure for determination of thermal denaturation midpoint Tm values for DmGSTD1, DmGSTD2, DmGSTD10, DmGSTE1, DmGSTE7, and DmGSTE14.

### 2.4. Protein Crystallization, Data Collection, and Structure Resolution

Crystallization conditions for DmGSTE1 were determined with an Oryx 8 robot (Douglas Instruments Ltd., Berkshire, UK) of the CRM2 crystallogenesis platform (University of Lorraine) using the sitting-drop vapor-diffusion method with purchased crystallization kits (Wizard™ Classic kits 1–4 from Rigaku Ltd. (Tokyo, Japan), Structure Screens 1–2 from Molecular Dimension Ltd., Classic kits 1–10 and JCSG kit from JENA Bioscience Ltd. (Thuringia, Germany), 624 conditions). Then, 50 µL of the crystallization solutions were deposited in the reservoirs of 96-well plates (MRC 2 Lens Crystallization Plate). Drops were prepared by mixing 0.3 µL of the protein solution and 0.3 µL of the crystallization solution, which were deposited in the wells of the 96-well plates. The protein solution contained 15 mg.mL^−1^ DmGSTE1 in 20 mM Tris-HCl, pH 7.4. Crystallization plates were stored at 4 °C. Sizable crystals for X-ray experiments were obtained with crystallization condition Wiz 2–28, which contains 20% *w*/*v* PEG 8000, 100 mM MES pH 6.0, and 200 mM Ca acetate.

Prior to the X-ray experiments, the crystals were quickly soaked in their mother liquor supplemented with 20% glycerol and flash-frozen at 100 K in a nitrogen stream. The diffraction data were collected on beamlines PROXIMA-2 at SOLEIL synchrotron (Saclay, France). The data were indexed and integrated with XDS [23], scaled, and merged with Aimless from the CCP4 suite [24]. The structure of DmGSTE1 was solved using the BALBES automated molecular replacement server [25]. The crystal structure of GSTE2 from *Anopheles gambiae* presenting 42.5% of sequence identity (PDB entry 2IMI) served as the best template. The DmGSTE1 structure was then refined with Refmac5 [26] and manually adjusted with COOT Coot [27]. Validation of all structures was performed with MolProbity [28] and the wwPDB validation server (http://validate.wwpdb.org, accessed on 18 March 2024). Crystal data and diffraction and refinement statistics are shown in Table 1, and all structural figures were generated with PyMol. Coordinates and structure factors were deposited in the Protein Data Bank (PDB entry 9F7K).

### 2.5. Bioinformatic Analysis of Structural Features at the Dimerization Interfaces

The amino acid GST sequences were downloaded from the uniprot website (https://www.uniprot.org/, accessed on 15 March 2024). Sequences were aligned using BioEdit software (https://bioedit.software.informer.com/ accessed on 15 March 2024). The features of the dimerization interfaces of each GST, namely, the number of total residues at the interface, the interface areas, the disulfide bonds, the number of hydrogen bonds, and the number of salt bridges, were computed with the online tool PDBePISA (https://www.ebi.ac.uk/pdbe/pisa/ accessed on 15 March 2024). Calculation was performed for the crystal structure of DmGSTE1 determined in this study and for each GST structure determined previously: DmGSTD1 (pdb 3EIN), DmGSTD2 (pdb 5F0G), DmGSTD10 (pdb 3F6F), AmGSTD1 (pdb 8Q89), DmGSTE6 (pdb 4YH2), DmGSTE7 (pdb 4PNG), and DmGSTE14 (pdb 6T2T).

## 3. Results

### 3.1. Sequence Analysis of Drosophila GST Epsilon and GST Delta

Delta and Epsilon Drosophila GSTs exhibit shared characteristics in their amino acid sequences. The glutathione binding site is comprised of residues from the thioredoxin fold, including the conserved cis-Pro located in strand β3, as well as the N-terminal two residues of helix α3 (Glu and Ser). Additionally, it features a histidine residue from a conserved β-turn connecting helix α2 and strand β3. Secondly, both classes show conservation in helix α3 and helix α4 due to their significance in dimerization. The “Clasp” motif is observed in the Delta class and the “Wafer” motif in the Epsilon class. The Clasp motif, also named lock-and-key ‘’Clasp’’ motif, has two main residues in each subunit located within helix α4. In the case of DmGSTD2, Tyr103 is involved in the ‘key’ and Met106 is involved in the ‘lock’ (Figure 1). Analysis of the wafer dimerization interface revealed a more complex motif. According to Wongsantichon 2015, the characteristic wafer motif of GSTEs is composed of an extended ‘wafer’ arrangement of two histidines (His71 and 103 in DmGSTE7) from helices 3 and 4, which is supported by interactions with two conserved serines from helices α4 and α6 (Ser106 and 164, respectively), as shown in Figure 1 for DmGSTE7 [29]. The alignment of the sequences of Drosophila GSTs indicates some variation in the nature of the hydrophobic residues involved in the dimer stabilization through the hydrophobic effect in both clasp and wafer motifs with majority Tyr or Phe residues for Delta GSTs and His residues for Epsilon GSTs (Figure 2). DmGSTE1 and DmGSTE2 lack histidine (position 103) but have a phenylalanine residue at the clasp position, similar to GSTDs. Concurrently, the adjacent serine residue that is part of the wafer motif in canonical GSTEs is replaced by alanine. DmGSTE1 and DmGSTE2 were previously shown to be closely related from an evolutionary point of view, supporting these common features between both GSTs [9]. The crystal structures of the DmGSTE1 orthologue in *Drosophila mojavensis* (DmojGSTE1) and *Musca domestica* (MdomGSTE7) were deposited in the Protein Data Bank [30]. Our alignment revealed that these structures presented an original Clasp + Wafer interface. Indeed, the sequence analysis showed differences in the interface motif of both the sequences compared to the DmGSTE1 (Figure 2). Both presented a tyrosine residue at position 103 (DmGSTE1 numbering), as in the Clasp motif from the Delta GSTs, whereas DmGSTE1 possessed a phenylalanine residue. Serine at position 106 found in the “wafer” motif is conserved in DmojGSTE1, unlike in DmGSTE1 and MdomGSTE7, where it is replaced by an alanine residue. This indicates that DmGSTE1 possesses a different and original motif compared to other Drosophila GSTs, as well as being different from the structures corresponding to similar sequences found in other species (50.9% identity with DmojGSTE1 and 52.6% with MdomGSTE7).

### 3.2. Enzymatic Characterization

In order to examine the general enzymatic competence of DmGSTE1, the classical CDNB substrate was tested with success. The glutathione conjugation of CDNB showed a k_cat_ of 234 min^−1^, with a K_M_ of 15 ± 4 µM for CDNB and a K_M_ of 226 ± 35 µM for GSH. The efficiency of 260 s^−1^.mM^−1^ appears similar to the 230 s^−1^.mM^−1^ previously measured for DmGSTE1 [31]. Notably, this efficiency is the numerical average of other Epsilon insects and is in close proximity to the two highest performing efficiencies measured for DmGSTD11 and *Apis mellifera* GSTD1 (respectively, 281 s^−1^·mM^−1^ and 320 s^−1^·mM^−1^) [20,31,32]. Interestingly, DmGSTE1 has a cysteine at position 16 close to the catalytic serine at position 14 according to our alignment (Figure 2). The Cys residue located within the active site can potentially catalyze a thiol transferase activity, and the activity was revealed using a glutathione reductase-coupled method following the consumption of NADPH. An activity of 0.5 ± 0.2 s^−1^ was measured with 200 µM of the hydroxyl-ethyl disulfide (HED) spontaneously reacting with the reduced glutathione during substrate turnover. These observations show similar activity measurements under the same conditions at a saturating concentration of HED for the rat and human GSTO1 (k_cat_ of 1.2 ± 0.1 and 1.6 ± 0.1 s^−1^, respectively) with both lacking intrinsic GSH transfer activity behavior with the CDNB substrate [33,34]. However, the turnover rate is depreciably lower in comparison to the recorded k_cat_ value of 10^2^ s^−1^ measured for the best thiol transferase activity for a known glutathione transferase—the dehydroascorbate reductase (DHAR) family, a class of GST restricted to the green lineage [12].

### 3.3. DmGSTE1 Crystal Structure

The crystal structure of DmGSTE1 was solved at 1.80 Å resolution in the *P*2_1_2_1_2_1_ space group. The asymmetric unit contains one dimer of DmGSTE1. Each subunit adopts the typical GST fold with an N-terminal thioredoxin domain (β1α1β2α2β3β4α3, Ser2-Tyr88) and an all-helical C-terminal domain (α4α5α6α7α8, Pro89-Asp223) (Figure 3A). The active site of DmGSTE1 is situated in a cleft between both domains. The glutathione binding site (G site) is occupied by a mixture of glutathione and glutathione sulfoxide refined with equal occupancies. The glutathione is stabilized by polar interactions with the main chain atoms of Val57 and Pro58 (cysteinyl moiety) and the sidechains of His43 and His55 (carboxy terminus) and Asp69 and Ser70 (gamma-glutamyl moiety). The sulfur atom of the reduced form of glutathione is oriented towards the solvent, while the sulfoxide group of the oxidized form establishes a hydrogen bond with the Ser14 sidechain (Figure 3B). This Ser14 is part of the catalytic motif ‘SPCV’ at the N-terminal end of helix α1 of DmGSTE1. Interestingly, the Cys16 residue part of this same catalytic motif is oriented toward the solvent and could be responsible for the thiol transferase activity observed with HED, as observed previously for other Ser-GSTs harboring a Cys in the catalytic motif [35]. No precise localization of the hydrophobic site (H site) could be made because of the absence of bound substrates, but this H site is probably situated in the hydrophobic cleft formed by helices α4, α6, and α8 near the G site.

As expected, a search for homologous structures in the PDB carried out with PDBeFold [35] resulted in the retrieval of GST structures of the Epsilon class. A structural comparison was performed of all Drosophila GST Epsilon structures available at the time of the study: DmGSTE6 (pdb 4PNF), DmGSTE7 (pdb 4PNG), and DmGSTE14 (pdb 6T2T). Structural superimposition of the corresponding monomers onto the DmGSTE1 monomer indicated high structural homology represented by an rmsd below 1 Å. Structural superimposition of the corresponding dimers onto the DmGSTE1 dimer resulted in an rmsd of 1.32 Å and a sequence identity of 49.8% (DmGSTE1 versus DmGSTE6), 1.22 Å and 49.8% (DmGSTE1 versus DmGSTE7), and 2.33 Å and 30.3% (DmGSTE1 versus DmGSTE14). For this last paralog, a higher rmsd value could be explained by large differences at regions involved in the dimerization interface, notably helices α4 and α5. DmGSTE14 was previously characterized and proved to exhibit a very atypical dimerization with multiple contacts between subunits [19].

We made a thorough analysis of the DmGSTE1 dimerization interface to have a tridimensional understanding of the variations in the wafer motif observed from the sequence analysis. The clasp residue around the axis of the dimer was Phe103, as predicted from the sequence alignment. Both Phe103A and Phe103B interacted by Π-Π stacking, similar to GST Delta, while most GST Epsilon structures had a His. The presence of a Phe instead of a His in terms of subunit communication by opposition to a classical GST Epsilon remains to be explored. The pair of Phe103 was bordered by the His71 residues from chains A and B, located in the active sites of both subunits. With a distance of around 4 Å between the side chains of Phe103 and His71, a potential pi-pi stacking could be formed (and could add to the first between Phe103) even if the planarity is not perfect between both aromatic rings (Figure 3C,D). This could be due to the interaction between glutathione and His71, as the residue at this position in GST Epsilon was shown to move upon binding to glutathione. This residue was also suggested to be crucial for communication between both subunits’ active sites [29]. Additional interactions around the dimer axis also involved the side chains of Ser107, Ala111 through hydrophobic interactions and salt bridges between Arg95 and Asp79 (intra-subunit, albeit stabilizing the global assembly), and Glu131 and Lys132. Additionally, several residues like Tyr88, Asn99, Asn115, and Glu131 established water-mediated hydrogen bonds between subunits. In summary, the overall assembly was stabilized by a combination of polar interactions surrounding the hydrophobic core of the dimer with the clasp Phe103 as a central and crucial anchor point. This makes GSTE1 similar to the classical Epsilon GSTs with the exception of the clasp residue being a Phe in GSTE1 instead of a His. This makes the motif of GSTE1 a hybrid type between clasp interfaces of GST Delta and wafer interfaces of GST Epsilon. As a consequence, we propose to name the interface ‘wafer/clasp’ in GSTE1 in reference to both types.

### 3.4. Thermal Stabilities and Structural Features of Interfaces in Drosophila GST Epsilon and Delta

As mentioned in the introduction, the dimerization interface plays a crucial role in the stability of GST structures. To evaluate and identify the key features responsible for this stability, we produced and purified all the GSTs from *D. melanogaster* for which the X-ray structure was previously solved. This allowed us to determine the thermal stability of each enzyme by monitoring the thermal stability through changes in secondary structure using the dichroic signal of each protein during a temperature increase (Table 2 and Appendix A). Additionally, based on the X-ray structures previously solved or solved in this study for GSTE1, we calculated the interface area and the total number of residues forming the interface, as well as the number of hydrogen bonds, salt bridges between both monomers, as well as the disulfide bonds (Table 2).

DmGSTE6 and DmGSTE7 presented similar features in terms of interface type and all other parameters, allowing us to describe their interface; consequently, it was not surprising to measure a similar Tm for both these enzymes. DmGSTE1, which presented a Clasp-Wafer interface, also had the larger interface area with approximately 100 Å^2^ additionally to DmGSTE6 and DmGSTE7. Even if the number of hydrogen bonds or salt bridges was inferior, due to this higher contact surface, DmGSTE1 presented a higher stability. DmGSTE14, which presented a higher interface surface compared to DmGSTE6 and DmGSTE7 (but lower compared to DmGSTE1), aided by a higher number of hydrogen bonds and salt bridges compared to all other Epsilon DmGSTs, presented the highest Tm, slightly higher compared to GSTE1. In general, all Delta GSTs presented a much lower Tm despite a contact surface of the same order and a similar number of interacting residues. This difference can be explained by the interface, indicating that the clasp stabilizes the Delta structures to a lesser extent. Interestingly, AmGSTD1 appeared with the lowest interface area (one-third less compared to all others) but with the highest stability compared to all other tested Delta GSTs, which appeared similar to the stability measured for the Epsilon class, probably due to the sulfur sandwich interface constituted by a disulfide bridge. It should be noted that the size of the dimerization interface and the nature of the interactions do not explain the increased stability of DmGSTD1, which appeared higher than the other two GSTs that presented the clasp interface without the sulfur sandwich. Additionally, it cannot be excluded that the loss of the dimeric state occurred at a lower temperature compared to the loss of a secondary structure. In conclusion, in a general manner, the interface type appears to be one of the main drivers of GST stability. The original Clasp-Wafer interface motif can confer an evolutionary advantage, supporting higher stability for DmGSTE1.

## 4. Discussion

Among the Delta and Epsilon classes, 22 GSTs were structurally described (Table 3). Different types of dimerization interfaces were observed and formalized for some of them in this study, as shown in Table 3. The Wafer and Clasp motifs were the most redundant dimerization interfaces observed within the structurally solved GSTs. The Clasp interface appeared mainly in the Delta class and the Wafer interface appeared mainly in the Epsilon class. However, our analysis revealed different additional types of interfaces. The Clasp + Wafer described in this study for DmGSTE1 was also observed for two other Epsilon GSTs from other insect species. Also, original dimerization interfaces have been observed in single structures to date. The X-ray structure of GSTE14 revealed a unique and original dimerization interface different to the Wafer motif expected for the Epsilon. Concerning DmGSTE14, in addition to the “multi-contact” motif, a Cys residue was observed in place of the Wafer motif, likely facilitating the establishment of a disulfide bond between both subunits forming the dimer as observed in the “sulfur sandwich” interface dimerization of the GSTD1 from *Apis mellifera* [8,19]. Additionally, of the multi-contact or the sulfur sandwich already described, an original interface was observed in the AdGSTD5. Indeed, the organization of the AdGSTD5 interface corresponded to a partial wafer motif: the Tyr103 and the Met106 generally found with the Clasp were now substituted by a His and Leu, respectively. We propose to name this motif “half Wafer”. BmGSTD1 presented also a substitution at the same position of the Clasp motif but, in this particular case, by an Asn at position 103 and also a Leu at position 106. Consequently, we propose to name this Delta interface based on polar contacts “polar” interface. Further, to date, seven different types of dimerization interfaces have been observed in insect Delta and Epsilon GSTs, as shown in the Table 3. A structural similarity analysis performed using DALI [37] on all solved structures of GST Epsilon and Delta to date allowed us to visualize the distribution of these atypical interfaces across the two classes, Epsilon and Delta (Figure 4).

Concerning *D. melanogaster*, most of the GSTs presented the Clasp and Wafer motifs for the Delta and the Epsilon GSTs, respectively. Four different types of dimerization motifs have been observed to date in *D. melanogaster* GSTs: Clasp, Wafer, Multi-contact, and Clasp + Wafer with the structure of DmGSTE1 from this study. This new interface within the Drosophila GSTs revealed by this study is close to the previously observed Clasp + Wafer already observed for DmojGSTE1 and MdomGSTE7. It indicates that this motif is conserved across the insect species as observed for the Clasp and Wafer. The structural similarity analysis revealed that these three paralogs appear on the same cluster, suggesting highly conserved structural features (Figure 4). The Clasp + Wafer dimerization interface found in DmGSTE1 is associated with a high thermal stability in a general manner compared to Delta and Epsilon GSTs and more particularly within the Epsilon GST. The high number of GSTs in both Epsilon and Delta classes contributes to functional redundancy, likely favoring greater diversity in terms of evolution. Interestingly the type of dimerization interface is directly related to the evolutionary distance between the different GSTs found in *D. melanogaster*. Phylogenic analyses of the *D. melanogaster* Delta and Epsilon GST protein sequences revealed two separate groups, one for each class, each presenting preferentially the Clasp and Wafer as observed in a general manner for insects [9]. For *D. melanogaster*, further details revealed that the Delta classes appeared organized into two main subgroups, one formed by DmGSTD11A and DmGSTD11B and the second formed by all others. DmGSTD1, DmGSTD2, and DmGSTD10 all included in a more recent group presented the same Clasp motif. Epsilon GSTs were organized into three subgroups: one resulting from an ancient separation, which includes DmGSTE14 exhibiting the original multi-polar interface. The other two subgroups encompass all other Epsilon GSTs, which are more closely related due to a more recent diversification presenting a more similar interface. DmGSTE6 and DmGSTE7 both belong to the same subgroup and exhibited the Wafer interface. GSTE1, on the other hand, belongs to another subgroup and presented the Clasp + Wafer interface, which is more similar to the Wafer motif compared to the multi-polar interfaces. Sequence alignment revealed that DmGSTE2 harbors the same residues involved in the motif as DmGSTE1, suggesting a Clasp + Wafer motif for DmGSTE2, as found in this study for DmGSTE1. The Clasp + Wafer motif is compatible with a good catalytic efficiency toward the CNDB in the average of other Epsilon GSTs. Additionally, the thiol transferase activity associated with DmGSTE1 is likely permitted by the Cys16. GSTs including insect GSTs present allosteric behavior for some substrates. The dimerization interface is key to explaining the allosteric phenomenon [16]. Future studies exploring the relationship between the interfaces and the allosteric behavior of insect GSTs represent an open avenue of research, now made possible by recent explorations into the diversity of dimerization interfaces. Additionally, along with allostery, another highly probable evolutionary driver of the dimerization interface is GST thermal stability. Our thermal stability analysis supports the correlation between the dimerization interface and stability. The type of interface appears to have a stronger influence on stability compared to the number of hydrophilic inter-subunit interactions. The type of interface probably correlates to the area of the interface explaining the larger area in the case of DmGSTE1 than supporting arguments for higher stability. Additionally, a low stability link to a low interface area can be compensated by a covalent link between both subunits, as observed in the case of AmGSTD1, which presents a disulfide bond between both monomers. The diversity of dimerization interfaces likely supports a range of functions linked to the activity and allosteric properties of each GST, while simultaneously adapting these interfaces to maintain their thermal stability. This work presenting a Clasp + Wafer interface for *D. melanogaster* in conjunction with two already observed for Epsilon GSTs explored the relationship between the interface and the thermal stability in *D. melanogaster*, suggesting some key functional features including the disulfide bond or interface area provoking new areas of discovery for insect-based catalysis.

In conclusion, in this study, the compilation and formalization of all different interfaces based on all known Delta and Epsilon GST X-ray structures allow us to propose seven different interfaces for these insect GSTs. This number is expected to increase in the future and will not only enable the emergence of broader studies on the link with stability based on this initial study but also likely open exploration of the role of these interfaces in their catalytic activity and, more specifically, their allosteric properties.

## Figures and Tables

**Figure 1 biomolecules-14-00758-f001:**
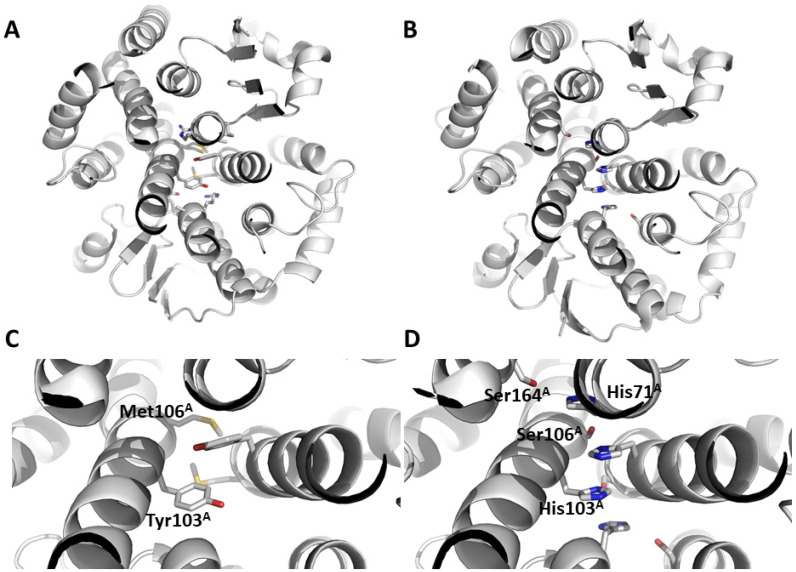
Clasp and Wafer structural motifs. The amino acids involved in the structural motifs “Clasp” (**A**,**C**) and “Wafer” (**B**,**D**) are indicated in stick mode in the general view (**A**,**B**) and in the focused view (**C**,**D**). The PDB codes 5F0G and 4PNG corresponding, respectively, to the DmGSTD2 and DmGSTE7 were used to exemplify the Clasp and Wafer interface dimerization motifs. The names and numbering of the amino acids are indicated only for one subunit (chain (**A**)) for clarity.

**Figure 2 biomolecules-14-00758-f002:**
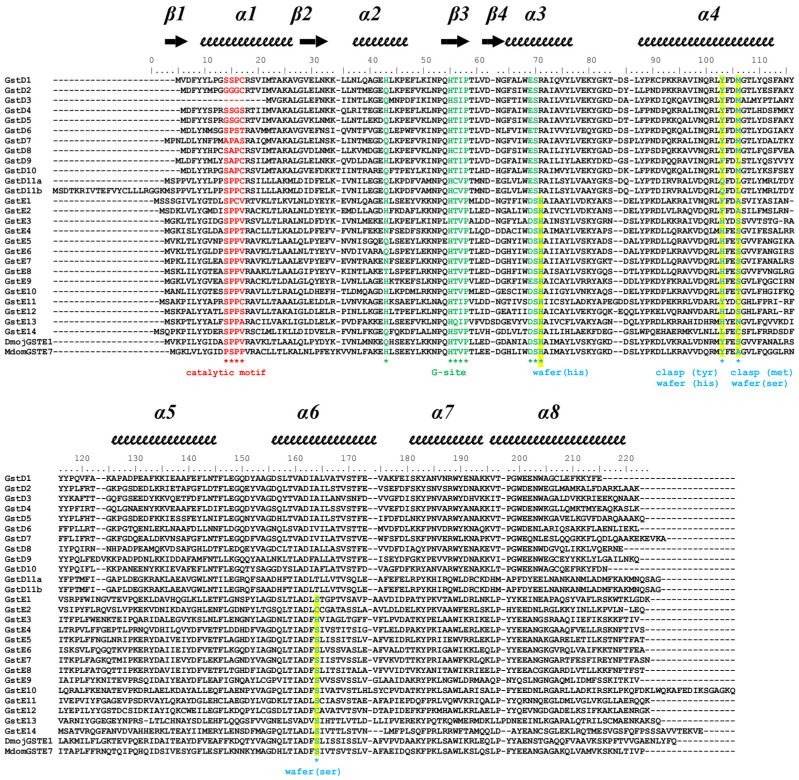
Sequence alignment of Delta and Epsilon GSTs**.** All known Delta and Epsilon GSTs from *Drosophila melanogaster*. The GSTE1 from *Drosophila mojavensis* and GSTE7 from *Musca domestica* are also represented. The numbering was based on the DmGSTE1 sequence. Secondary structures were inferred from the X-ray structure of DmGSTD2 (5F0G [21]) and are reported above the alignment. Conserved sites are annotated below the alignment.

**Figure 3 biomolecules-14-00758-f003:**
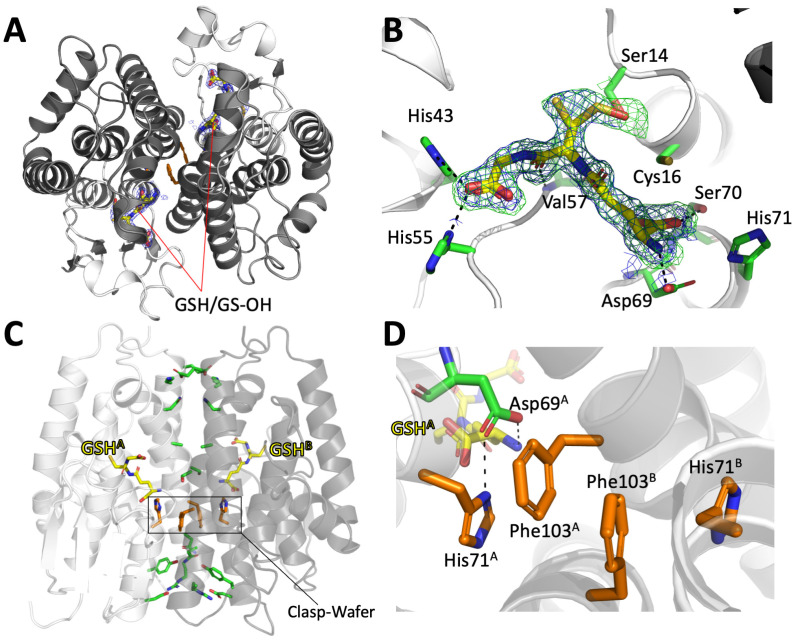
Crystal structure of DmGSTE1 in complex with glutathione. (**A**) Overall view of DmGSTE1 dimer with each active site occupied by a glutathione molecule. N-terminal domains are colored white and C-terminal domains are colored grey. (**B**) Enhanced view of the glutathione binding site (residues as green sticks) bound to glutathione (yellow sticks). The 2mFo-DFc electron density map is contoured at 1.2 sigma in blue. The mFo-DFc omit map is shown at 3.0 sigma in green. (**C**) Dimerization interface residues are colored green, the atypical clasp-wafer motif is colored orange. Only reduced GSH is shown for clarity. (**D**) Enhanced view of the clasp-wafer motif in polar contact with the adjacent Asp69 residue from the glutathione binding site. Only reduced GSH is shown for clarity.

**Figure 4 biomolecules-14-00758-f004:**
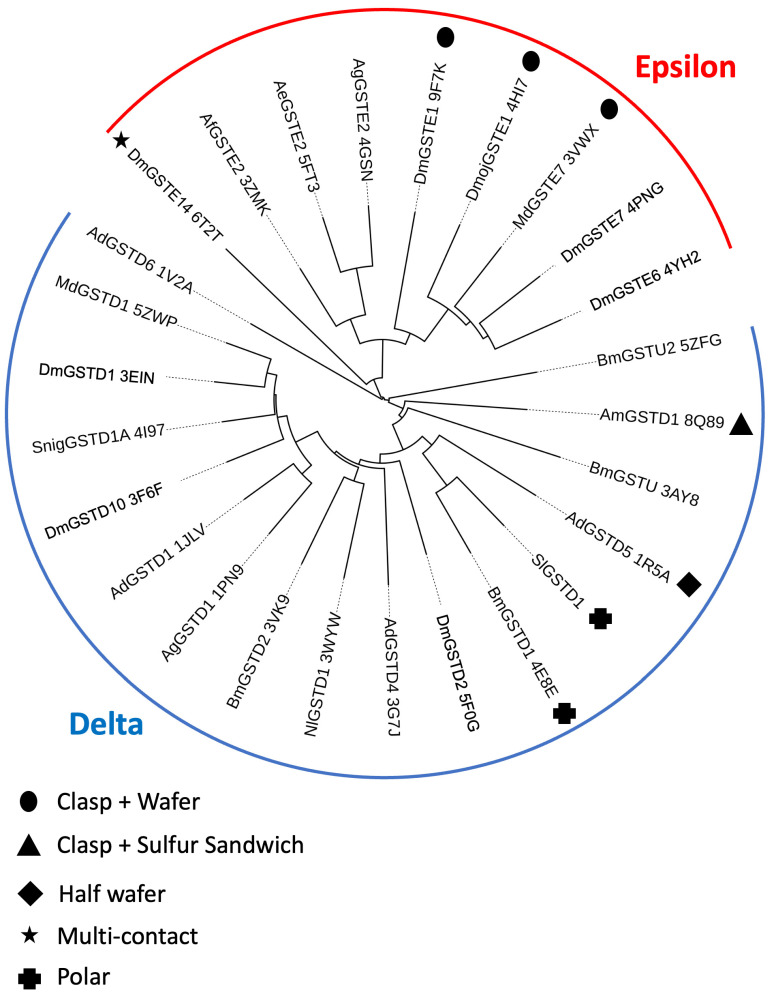
Circular dendrogram of Delta and Epsilon GSTs. The structural alignment and the corresponding structural similarity tree were calculated with the ‘all-against-all’ option implemented on the DALI webserver (http://ekhidna2.biocenter.helsinki.fi/dali/, accessed on 10 March 2024) [37]. The tree was edited with iTOL (https://itol.embl.de, accessed on 10 March 2024). The PDB codes are provided for each structure. Two main branches correspond to the GST Delta (blue) and the GST Epsilon (red). The atypical interface motifs are depicted with black pictures.

**Table 1 biomolecules-14-00758-t001:** Statistics of X-ray diffraction data collection and model refinement. ^1^ Values in parentheses are for the highest resolution shell; Rmerge2=∑h∑iIhi−Ih/∑h∑iIh; Rmeas3=∑h∑inhnh−11/2Ihi−Ih/∑h∑iIh (with $I_{hi}$ the intensity of an individual observation of the reflection *h* and $\langle I_{h}\rangle$ its average of all symmetry-related or replicate observations); ^4^ *CC*_1/2_ is the correlation coefficient of the mean intensities between two random half-sets of data. Rwork5=∑hFo−Fc/∑hFo (95% of the reflections, *R_free_* same formula (5% of the reflections) ($F_{o}$ and $F_{c}$ observed and calculated structure factors, respectively). ^6^ Water molecules. ^7^ Glycerol molecules. ^8^ Glutathione molecules. ^9^ RMSZ: root mean square Z-score. ^10^ The MolProbity clash score is the number of serious clashes per 1000 atoms.

Data Collection	
Diffraction source	SOLEIL PROXIMA2
Wavelength (Å)	1.14071
Space group	*P*2_1_2_1_2_1_
*a*, *b*, *c* (Å)	81.04, 82.92, 95.54
*α*, *β*, *γ* (°)	90, 90, 90
Resolution range (Å)	41.46–1.80 (1.84–1.80) ^1^
Total number of measured intensities	394,908 (23,814) ^1^
Number of unique reflections	60,131 (3542) ^1^
Average redundancy	6.6 (6.7) ^1^
Mean *I/*sig(*I*)	17.5 (2.3) ^1^
Completeness (%)	99.2 (99.7) ^1^
*R*_merge_ ^2^; *R*_meas_ ^3^	0.054 (0.812) ^1^; 0.064 (0.961) ^1^
*CC*_1/2_ ^4^	1.00 (0.70) ^1^
Wilson *B*-factor (Å^2^)	26.5
**Refinement and structure**	
Resolution range (Å)	41.46–1.80
Number of reflections	60,131
*R_work_*/*R_free_* ^5^	0.1540/0.1770
Correlation *Fo*-*Fc*	0.97
Total number of atoms	7323
Number of non-protein molecules	208 HOH ^6^, 2 GOL ^7^, 2 GSH/GS8 ^8^
Average *B* factor (Å^2^)	31.0 (all), 30.5 (protein), 40.7 (waters), 46.1 (GOL), 33.3 (GSH/GS8)
**Model quality**	
RMSZ bond lengths ^9^	0.58
RMSZ bond angles ^9^	0.78
Ramachandran favored (%)	99.1
Ramachandran allowed (%)	0.7
Rotamer outliers (%)	0.2
Clash score ^10^	1

**Table 2 biomolecules-14-00758-t002:** Thermal stability and dimerization interface parameters. Each structurally solved Drosophila GST was analyzed using thermal denaturation, with secondary structure denaturation monitored using dichroic light signals. The melting temperature (Tm), corresponding to the half denaturation, was determined for each protein measured in this study, except for AmGSTD1, which was measured in a previous study under the same conditions [20]. For each structure, the interface area and the number of residues, hydrogen bonds, salt bridges, and disulfide bonds were calculated using the PISA software v 1.52 [36]. Regarding GSTE14, the presence of a disulfide bond was not definitively demonstrated (indicated by N.D.), as mentioned in the corresponding work detailing the resolution of this structure [19].

	Interface Type	Tm	Interface Area Å^2^	Number of Residues Involved in the Interface	Number of H-Bonds	Number of Salt Bridges	Number of Disulfide Bonds
DmGSTE1	Clasp + wafer	58.0 ± 0.1	1458	86	4	4	0
DmGSTE6	Wafer	56.0 ± 0.1	1348	79	5	0	0
DmGSTE7	Wafer	56.5 ± 0.1	1362	75	8	4	0
DmGSTE14	Multi-polar	58.3 ± 0.6	1431	76	19	6	N.D.
DmGSTD1	Clasp	53.0 ± 0.1	1395	72	4	0	0
DmGSTD2	Clasp	47.9 ± 0.1	1408	68	9	6	0
DmGSTD10	Clasp	44.7 ± 0.1	1410	70	4	0	0
AmGSTD1	Clasp + Sulfur sandwich	55.5 ± 0.6	1039	58	5	0	1

**Table 3 biomolecules-14-00758-t003:** Description of the interface type allowing the dimerization of GSTs in all experimentally solved insect GST structures.

CLASS	Insect Name	GST	PDB	Interface Type	Reference
**Delta**	*A. dirus*	AdGST1	1JLV	**Clasp**	[38]
		AdGSTD4	3G7J	Clasp	[39]
		AdGSTD5	1R5A	**Half Wafer**	[40]
		AdGSTD6	1V2A	Clasp	[40]
	*A. mellifera*	AmGSTD1	*8Q89*	**Sulfur sandwich**	[20]
	*A. gambiae*	AgGSTD1	1PN9	Clasp	[41]
	*B. mori*	BmGSTD1	4E8E	**Polar (Asn)**	N.P
		BmGSTD2	3VK9	Clasp	[42]
	*D. melanogaster*	DmGSTD1	3EIN	Clasp	N.P
		DmGSTD2	5F0G	Clasp	[21]
		DmGSTD10	3F6F	Clasp	[43]
	*M. domestica*	MdGSTD1	5ZWP	Clasp	[44]
	*N. lugens*	NlGSTD1	3WYW	Clasp	**[45]**
	*S. nigrita*	SnigGSTD1A	4I97	Clasp	N.P
**Epsilon**	*A. aegypti*	AeGSTE2	5FT3	Wafer	N.P
	*A. gambiae*	AgGSTE2	4GSN	Wafer	[46]
	*A. funestus*	AfGSTE2	3ZMK	Wafer	[47]
	*D. melanogaster*	DmGSTE1	9F7K	**Clasp + Wafer**	This study
		DmGSTE6	4YH2	Wafer	[29]
		DmGSTE7	4PNG	Wafer	[48]
		DmGSTE14	6T2T	**Multi-polar**	[19]
	*D. mojavensis*	DmojGSTE1	4HI7	**Clasp + Wafer**	*N.P*
	*M. domestica*	MdGSTE7	3VWX	**Clasp + Wafer**	**[30]**

All the interfaces that do not correspond to clasp or wafer are indicated in bold. The corresponding Protein Data Bank codes are indicated along with references when a publication supporting the solved structure exists. N.P indicates any related publication.

## Data Availability

All the data related to the structure are available on the PDB website (https://www.rcsb.org/).

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
