# Peer review of "Structural and Thermodynamic Insights into Dimerization Interfaces of Drosophila Glutathione Transferases"

_biomolecules, 2024, doi:10.3390/biom14070758_

Round 1

Reviewer 1 Report

Comments and Suggestions for Authors

In the current study, the dimerization interfaces of fly GSTs were comprehensively analyzed through sequence alignment, X-ray structure determination, and thermal stability measurement. Classification of the dimerization interfaces was proposed based on sequence alignment, structural analysis, and phylogenetic analysis. GST dimer stability was shown to be correlated with the dimerization interface structure. The work presented here provides deeper information on the structural and thermodynamic aspects of GST dimerization and therefore is a valuable addition to the field of research. However, there are a few unclear points. The comments to this paper are described below:

 1. Page 10, lines 353-356

  The thermostability seems considerably differ among the Delta GSTs. What features, such as structure within the Clasp interface or outside the interface, account for the difference in the thermostability?

 2. Page 11, lines 384-388

  Whether the disulfide bond is actually formed and contributes to the higher thermostability of DmGSTE14 can be estimated by comparing the thermostability in the presence and absence of DTT. I would recommend the authors to include such information if possible.

 3. Page 12, Figure 4

  Please keep the GST names and the PDB codes in the same order. For example: "3F6F DmGSTD10, AdGSTD1 1JLV" -> "DmGSTD10 3F6F, AdGSTD1 1JLV"

Author Response

Thank you for reviewing our article and for your insightful questions about our work.

  1. Page 10, lines 353-356

  The thermostability seems considerably differ among the Delta GSTs. What features, such as structure within the Clasp interface or outside the interface, account for the difference in the thermostability?

Among the delta class, AmGSTD1 exhibits increased stability due to the sulfur sandwich additionally to the clasp motif. Among the three other delta class GSTs with a clasp motif, DmGSTD1 appears significantly more stable than the two others. Neither the interface surface area nor specific interactions can explain this. A more in-depth approach using molecular dynamics tools might provide answers in the future. The following sentence was added in the text: “. It should be noted that the size of the dimerization interface and the nature of the interactions do not explain the increased stability of DmGSTD1, which appears higher than the other two GSTs that present the clasp interface without the sulfur sandwich.”

  1. Page 11, lines 384-388

  Whether the disulfide bond is actually formed and contributes to the higher thermostability of DmGSTE14 can be estimated by comparing the thermostability in the presence and absence of DTT. I would recommend the authors to include such information if possible.

It is challenging to reduce the disulfide bond in DmGSTE14, likely due to its buried nature. Even under denaturing conditions, when we attempt to reduce the disulfide bond and titrate the cysteines, we are unable to achieve their complete reduction. Consequently, obtaining a circular dichroism spectrum of the reduced form of the protein is difficult. Therefore, while comparing the thermostability in the presence and absence of DTT could provide some insights, the inherent difficulty in fully reducing the disulfide bond limits our ability to answer with the propose experiment.

  1. Page 12, Figure 4

  Please keep the GST names and the PDB codes in the same order. For example: "3F6F DmGSTD10, AdGSTD1 1JLV" -> "DmGSTD10 3F6F, AdGSTD1 1JLV"

Figure 4 was adapted consequently.

Reviewer 2 Report

Comments and Suggestions for Authors

In this paper, an analysis of the dimerization interfaces of fly GSTs through sequence alignment was performed. The X-ray crystal structure of GSTE1 was determined and provided insights into the variations within Delta and Epsilon GST interfaces. The dimeric state of GSTs is important for their function and, in several cases, the dimer formation supports allosteric properties including positive and negative cooperativity. 

The authors have carried out a very basic characterisation of the enzymes using CD spectroscopy. Other methods could have been used to provide a more convincing conclusion for the role of different interfaces. Seven different interfaces were proposed for insect GSTs but more detailed analysis is necessary for full understanding of these interfaces.

Comments for major improvement:

1. Provide the spectra obtained for each dichroism experiment (possibly as supplementary files). Explain how are the changes correlated with the interface and not with the overall stability of the protein. 

2. Could thermofluor be better to assess the stability?

3. Has kinetic characterisation and measurement of the activity been carried before and after thermal treatment?

4. Were the stability measurements performed in the presence or absence of GSH?

5. Provide sequence identity of the model used in molecular replacement and give also the seq. identities of the other GSTs (not only the rmsd)

6. Provide omit map for Fig. 3B. The electron density is quite weak at 1.2 sigma level. What is the temperature factor of GSH? Provide B-factors for protein atoms, GSH, waters. Give the number of water molecules in Table 1.

 7. Is glutathione or oxidised glutathione shown in Fig. 3B?

Author Response

Thank you for reviewing our article and for your insightful questions about our work.

  1. Provide the spectra obtained for each dichroism experiment (possibly as supplementary files). Explain how are the changes correlated with the interface and not with the overall stability of the protein. 

As requested a Supp figure 1 including all the spectra was added.

Recording the circular dichroism spectra of proteins allows for the measurement of secondary structure presence and its evolution with temperature. When secondary structure elements are lost, the interface is also lost. While we cannot exclude the possibility that the interface is less stable than the secondary structures and could be lost at a lower temperature, the strong correlation between the stability energies of the interface and the Tm values obtained in this study suggests that the two are indeed correlated. The following sentence was added to the text to indicate this possibility: 'Additionally, it cannot be excluded that the loss of the dimeric state occurs at a lower temperature compared to the loss of secondary structure.'"

  1. Could thermofluor be better to assess the stability?

Yes, this technique was primarily employed for screening substrates or inhibitors (doi: 10.1371/journal.pone.0137083). The method relies on a probe's ability to interact with hydrophobic areas of the protein. As demonstrated in this publication, the hydrophobicity of the interface area can vary, resulting in variable signals depending on the GST being tested and the nature of its interface. In this study, we opted to utilize the dichroic signal, which provides a curve for each temperature. We find this approach more straightforward to interpret compared to the single fluorescence signal measured at each temperature using the TSA method. However, it would be interesting to consider using this approach in future studies to complement the interface analyses initiated in this work.

  1. Has kinetic characterisation and measurement of the activity been carried before and after thermal treatment?

The kinetic measurements were conducted in a separate experiment. In this study, we did not assess the impact of heating on the enzyme activity. Additionally, circular dichroism required several hours of experimentation for each enzyme. It may be more appropriate to measure enzyme activity with a standardized duration at a single temperature.

  1. Were the stability measurements performed in the presence or absence of GSH?

All stability measurements were conducted in a buffer without GSH, as described in the Materials and Methods section. As observed previously for other GSTs using thermal shift assays, we can hypothesize that stability may vary in the presence of GSH. It cannot be ruled out, as observed previously, that adding GSH could either decrease or increase stability depending on the GST being tested, thereby influencing the parameters that explain differences between the GSTs under investigation. Furthermore, studying this effect in a future study would be of interest.

  1. Provide sequence identity of the model used in molecular replacement and give also the seq. identities of the other GSTs (not only the rmsd)

As requested the seq. identities was added for the GSTE2 from Anopheles gambiae used for the molecular replacement. As requested all the seq identies were indicated in addition of the RMSD value for the other comparison (line 304)

  1. Provide omit map for Fig. 3B. The electron density is quite weak at 1.2 sigma level. What is the temperature factor of GSH? Provide B-factors for protein atoms, GSH, waters. Give the number of water molecules in Table 1.

We have added a mFo-DFc omit map at 3.0 sigma, highlighted in green in Fig. 3B. The omit map clearly indicates the presence of the ligand within the structure. As requested, Table 1 has been completed with all the requested information.

  1. Is glutathione or oxidised glutathione shown in Fig. 3B?

We have indicated in Figure 3B which form of glutathione is observed; indeed, both oxidized and reduced forms are present in the structure.

Round 2

Reviewer 2 Report

Comments and Suggestions for Authors

The authors have sufficiently addressed all the comments.